# FtsZ induces membrane deformations via torsional stress upon GTP hydrolysis

Diego A. Ramirez-Diaz[1,2], Adrián Merino-Salomón[1,3], Fabian Meyer[4], Michael Heymann [1,5], Germán Rivas[6], Marc Bramkamp [4] & Petra Schwille [1✉]

FtsZ is a key component in bacterial cell division, being the primary protein of the presumably contractile Z ring. In vivo and in vitro, it shows two distinctive features that could so far, however, not be mechanistically linked: self-organization into directionally treadmilling vortices on solid supported membranes, and shape deformation of flexible liposomes. In cells, circumferential treadmilling of FtsZ was shown to recruit septum-building enzymes, but an active force production remains elusive. To gain mechanistic understanding of FtsZ dependent membrane deformations and constriction, we design an in vitro assay based on soft lipid tubes pulled from FtsZ decorated giant lipid vesicles (GUVs) by optical tweezers. FtsZ filaments actively transform these tubes into spring-like structures, where GTPase activity promotes spring compression. Operating the optical tweezers in lateral vibration mode and assigning spring constants to FtsZ coated tubes, the directional forces that FtsZ-YFP-mts rings exert upon GTP hydrolysis can be estimated to be in the pN range. They are sufficient to induce membrane budding with constricting necks on both, giant vesicles and *E.coli* cells devoid of their cell walls. We hypothesize that these forces result from torsional stress in a GTPase activity dependent manner.

[1] Department of Cellular and Molecular Biophysics, Max Planck Institute of Biochemistry, Martinsried, Germany. [2] Graduate School for Quantitative Biosciences (QBM), Ludwig-Maximillians-University, Munich, Germany. [3] International Max Planck Research School for Molecular Life Sciences (IMPRS-LS), Munich, Germany. [4] Institute of General Microbiology, Christian-Albrechts-Unversity, Kiel, Germany. [5] Institute of Biomaterials and Biomolecular Systems, University of Stuttgart, Stuttgart, Germany. [6] Centro de Investigaciones Biológicas Margarita Salas, Consejo Superior de Investigaciones Cientificas (CSIC), Madrid, Spain. ✉email: schwille@biochem.mpg.de

In biology, fundamental mechanical processes, such as cell division, require an intricate space-time coordination of respective functional elements. However, how these elements, mostly proteins, can self-organize to exert forces driving large-scale transformations is poorly understood. In several organisms, ring-like cytoskeletal elements appear upon cytokinesis; for instance, the FtsZ-based contractile Z ring in bacteria. Ring-like FtsZ structures have previously been shown to deform liposome membranes[1,2]. When reconstituted on flat membranes, FtsZ self-assembles into treadmilling vortices with conserved direction[3,4]. In vivo, FtsZ shows circumferential but bidirectional treadmilling that is assumed to serve as a pacemaker guiding peptidoglycan synthesis around the septum[5,6]. In vivo[5,6] and in vitro[3] experiments have suggested that the emergence of FtsZ treadmilling phenomenon is intimately related to its GTPase activity.

Despite of these exciting findings, it is not clear how exactly FtsZ treadmilling filaments may at all contribute to the physical process of constriction[7,8]. The challenge is twofold: (i) to determine the forces that are actually required to divide a bacterial cell with its much more complex architecture than a membrane shell, and (ii) to formulate the exact mechanism by which forces on membranes could at all be exerted by FtsZ treadmilling filaments. For instance, considering the mechanical bearing related to internal turgor pressure (~MPa), models have suggested that FtsZ forces in the range of 8–80 pN would be required for constriction[9]. In contrast, it has been proposed that turgor pressure need not be considered, due to the possibility of same

osmolarity between periplasm and cytoplasm[10]. For this case, very low FtsZ forces in the range of 0.35–2.45 pN could exert membrane deformations leading to constriction[10]. In conclusion, in vivo and in vitro experimental approaches addressing those two major questions are needed to gain deeper understanding in cell division in bacteria. While (i) can only be addressed by extensive in vivo studies and may remain a notorious challenge in bacterial cell biology for many years, (ii) is more readily accessible by state-of-the-art biophysics on in vitro reconstituted systems, aiming at elucidating the mechanistic features of FtsZ as a membrane-deforming polymer.

Regardless of whether or not FtsZ is the major force contributor in cell division, it remains a fundamental question whether forces can be associated to GTP consumption and whether there is a structural connection to the emergence of treadmilling. And if so, how this force could be transmitted to constrict the membrane. To approach these questions, we introduced an experimental strategy to produce cylindrical membrane geometries mimicking rod-like bacterial shapes, by pulling soft lipid tubes from deflated giant unilamellar vesicles (GUVs) using optical tweezers. Our aim is to quantitatively elucidate the physical principles underlying membrane deformations induced by dynamic FtsZ rings and the scale of delivered forces. These particular principles are key to understand the nature of FtsZ membrane deformations in vitro and in vivo.

## Results and discussion

Based on our recent study[3], we externally added FtsZ-YFP-mts to GUVs made of *E. coli* lipid extract. Conditions to obtain ring-like structures were determined by tuning GTP and Mg$^{+2}$ (Fig. 1a). Since no clear deformations were observed for tensed vesicles (Fig. 1a), we designed a two-side open chamber allowing for slow water evaporation to obtain deflated and deformable GUVs. After 20–30 min, we evidenced that rings were inducing inwards cone structures emerging from the membrane surface, indicative of drilling-like inward forces (Fig. 1b). Motivated by this specific geometry, we designed PDMS microstructures mimicking such inward cones (Figs. 1c and S1A). After coating these with supported lipid bilayer (SLB) and triggering protein polymerization, we observed individual filaments/bundles to wrap the cone in a dynamic fashion resembling a vortex (Fig. 1d) (Supplementary Movie S1). We noticed that the dynamic vortices rotate both clockwise and anticlockwise (Fig. 1e), indicating that preferential directionality observed on flat SLBs is absent in conical geometry. Rotational velocities were estimated around 43 nm/s, showing relatively good agreement with our previous results on flat surfaces (34 nm/s)[3].

To quantitatively characterize the impact of FtsZ on soft tubular geometries, we developed a method based on optical tweezers. Contrary to prior approaches using micropipettes[11], we pulled soft tubules from weakly surface-attached GUVs (Fig. 2a) by moving the GUVs relative to an optically trapped bead. Lipid tubes with mean diameter of ca. 0.47 μm (Fig. S2A) were now pulled from deflated GUVs decorated with ring-like FtsZ structures and inward-conical deformations (Supplementary Movie S2). Given the mobility of FtsZ rings and filaments over the GUV surface, protein from the vesicle started entering the tube immediately after pulling. After 175 s, helical tube shapes were clearly observed (Fig. 2b), indicative of dynamic coiling (Supplementary Movie S3). As more protein entered the tube, the spring-like structure became compressed (Fig. 2b, 500 s). These helical tube deformations can be rationalized by twisting of an elastic rod subjected to constant tensile force (Fig. 3f). Similar to the experiment in Fig. 1d, filaments grew toward (clockwise) and away from (counterclockwise) the tip of the tube. If filament

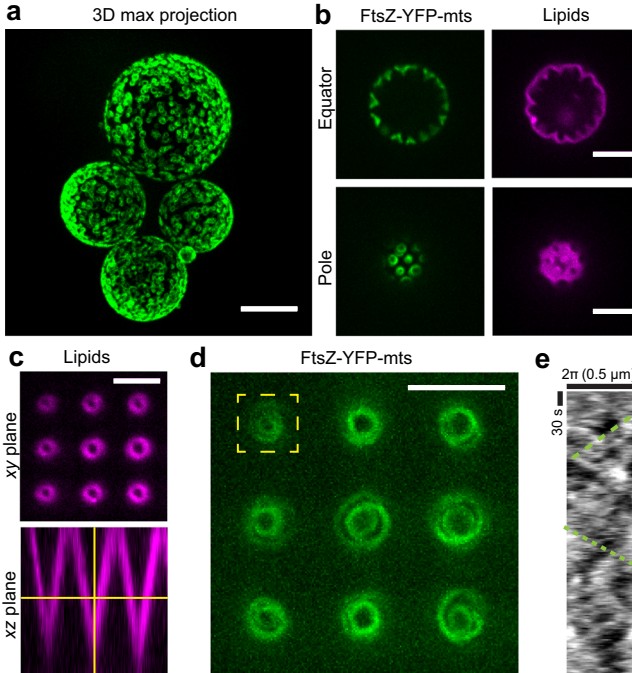

**Fig. 1 Bidirectional FtsZ treadmilling rings yield directional screw-like lipid membrane deformations. a** FtsZ-YFP-mts ring structures externally decorating GUVs (scale bar = 10 μm). **b** After GUV deflation, inwards conical deformations emerged from FtsZ rings (GUVs: N > 20) (scale bar = 5 μm). **c** Inspired by deformations in (**b**), we designed a PDMS microstructure with inwards conical geometry covered with a supported lipid bilayer (SLB). The imaging plane was chosen to have a cross-section of 1–2-μm diameter (scale bar = 5 μm). **d** Inside cones, FtsZ-YFP-mts self-assembled into dynamic vortices (Supplementary Movie S1) (scale bar = 5 μm). **e** A representative kymograph (N > 3) showed negative and positive slopes indicating the presence of clockwise and anticlockwise directions.

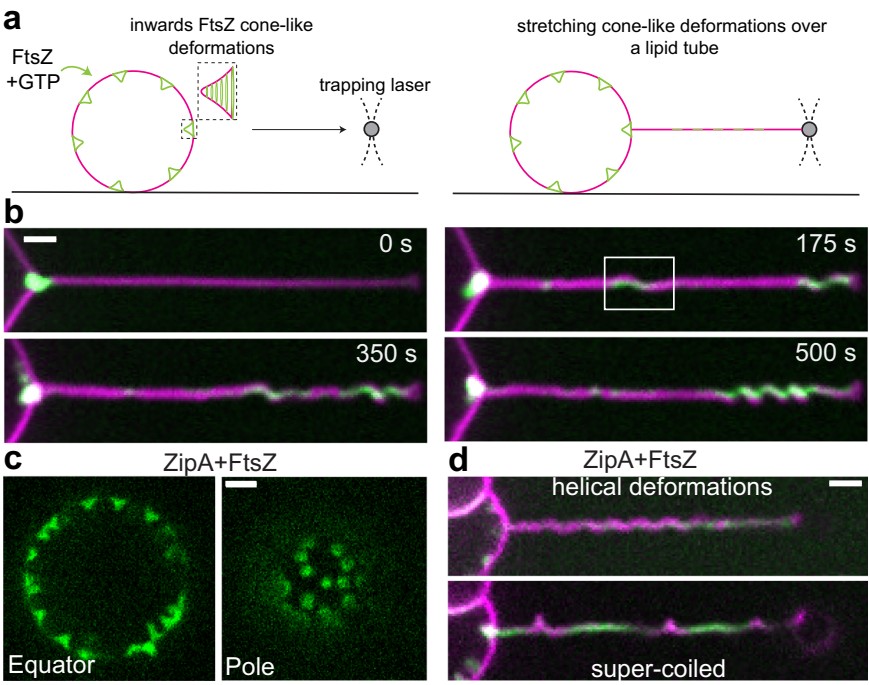

**Fig. 2 Inwards FtsZ-induced conical deformations are geometrically equivalent to spring-like deformations on a tubular surface. a** To characterize the protein structures forming the cone-like deformations, we stretched the deformed vesicle membrane into a tubular geometry. Large and soft lipid tubes were pulled from weakly surface-attached GUVs by moving the GUVs relative to an optically trapped bead. **b** A representative time acquisition shows that FtsZ-YFP-mts entered the tube promoting coiling as a function of time ($N = 12$). Clustering of protein toward the tip correlates with a spring-like shape. Green and magenta corresponds to fluorescence signal of FtsZ-YFP-mts and lipid, respectively. **c** To rule out that artificial attachment of the FtsZ-YFP-mts is responsible of the helical transformation, we reconstituted wild-type FtsZ anchored to the membrane via ZipA. When wild-type FtsZ was added to ZipA-decorated vesicles, rings were self-assembled causing inwards cone-like deformations ($N = 74$) such as in the case of FtsZ-YFP-mts. **d** After pulling lipid tubes, similar helical deformations and supercoiled regions were observed confirming that torsion is related to the FtsZ core of the polymer ($N = 3$ out of 18 pulled tubes). Fluorescence signal of wt-FtsZ-Alexa 488 is shown in green while siZipA remains unlabeled and lipids are shown in magenta. (Scale bar = 2 µm).

growth imposes torsion, the counter-growing filament will generate torsion in the opposite direction. The importance of bidirectional filament growth can be understood using a shoelace analogy: opposite torque should be exerted on both ends of the shoelace to observe a helical deformation. If one end is loose, the opposite end will only rotate accordingly (sliding).

Since spring-like deformations were observed with a FtsZ protein chimera that binds autonomously to membrane (FtsZ-YFP-mts), we attempted to confirm whether this phenomenology is intrinsic to the FtsZ polymer and not due to chimera artifacts, e.g., induced by the membrane targeting sequence. Based on the reconstitution of treadmilling dynamic rings on flat membranes using the *E. coli* FtsZ natural anchor ZipA[12], we aimed to establish appropriate conditions for WT-FtsZ rings externally decorating GUVs using ZipA. First, as a control, ZipA-decorated vesicles were examined under deflation conditions, and none of them showed inwards deformations ($N = 14$, Fig. S2B). In addition, we pulled lipid tubes with only ZipA and none showed any relevant deformation over time ($N = 10$, Fig. S2B). Only after adding wild-type FtsZ, we obtained rings and inward deformations (Fig. 2b). We then pulled tubes from these vesicles and observed helical transformations (Fig. 2c) ($N = 3$ out of 18 pulled tubes), indicating that FtsZ polymer and not its membrane attachment (in this case ZipA) caused this effect. Interestingly, FtsZ+ZipA (as well as FtsZ-YFP-mts) displayed in plectonic/supercoiled regions (Figs. 2c and S1E) as further indicative of torsion over the lipid tube.

After having established that the spring-like membrane transformations do not result from membrane anchors only, we

needed to explore the active role of the GTPase activity. To investigate the role of GTP hydrolysis for the spring-like deformations, we reconstituted FtsZ-YFP-mts*[T108A], a mutant with low GTPase activity[3]. This mutant self-assembles into rings at similar times (comparable polymerization rates) with respect to FtsZ-YFP-mts but lacks dynamic treadmilling[3]. Once reconstituted here on soft vesicle surfaces, FtsZ-YFP-mts*[T108A] rings (Fig. S1B) also induce cone-like deformations (Fig. S1C) as well as helical deformations, after 300 s, in lipid tubes (Fig. 3b). The pitch of these helices, however, remained unaffected ($\lambda > 3$ µm) at longer times (900 s, Supplementary Movie S4). In contrast, helices decorated with GTP-active FtsZ-YFP-mts (Fig. 3a) underwent compression to a pitch of $\lambda \sim 1.5$ µm already before 300 s. By plotting the arclength of the spring against FtsZ density on the tubes in Fig. 3a, b as a function of time (Fig. S2D), we clearly observed a greater membrane-deforming activity for FtsZ-YFP-mts (Fig. 3c). The timescale of deformations likely depends on the amount of protein on the tube. Thus, since the initial amount of protein on the tubes varies among independent experiments, further statistical analysis was performed in steady state.

Since the deflation of individual GUVs could fluctuate, we also tested whether compression could be biased by GUV membrane tension and protein density over the tube. The tube diameter $d$ represented our observable for membrane tension according to the relation $d = \sqrt{\frac{2\kappa}{\sigma}}$, where $\kappa$ denotes the lipid bending modulus and $\sigma$ the membrane tension[11,13]. The lower the membrane tension by deflation, the larger the tube diameter. Therefore, we plotted the mean pitch vs. tube diameter (Fig. 3d), considering

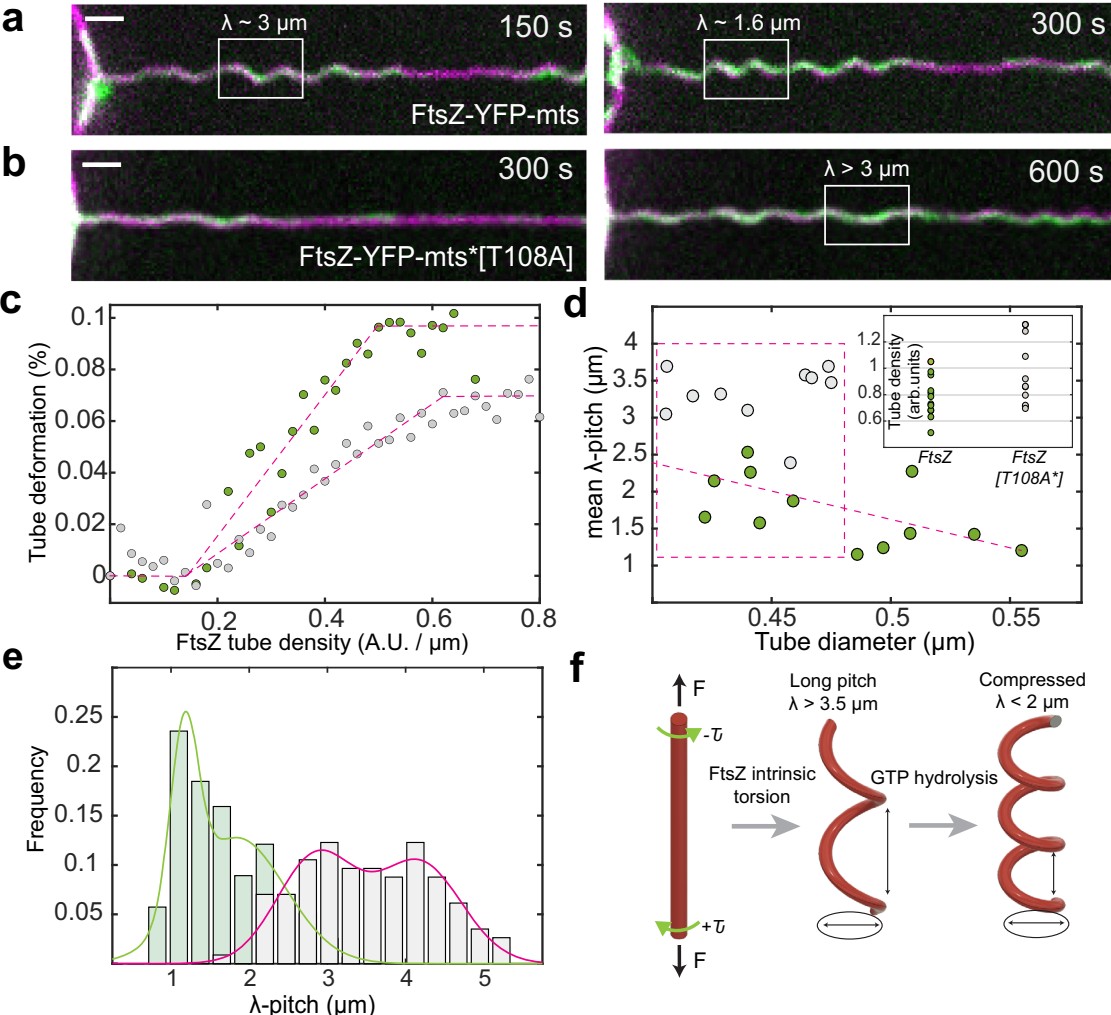

**Fig. 3 GTPase activity in FtsZ filaments promote spring compression and condensation.** Time-lapse acquisition of lipid tubes covered by **a** FtsZ-YFP-mts ($N = 12$) and **b** FtsZ-YFP-mts*[T108A] ($N = 10$). Both proteins promote helical deformations with the difference that GTPase activity induces compression ($\lambda \sim 1.6\,\mu m$) of initially longer pitch ($\lambda > 3\,\mu m$). FtsZ-YFP-mts is shown in green while lipids are shown in magenta (scale bar = 2 μm). **c** Tube deformation (arclength) in (**a, b**) against FtsZ-YFP-mts (green circles) and FtsZ-YFP-mts*[T108A] (gray circles) tube density, as function of time (Fig. 2SD), evidenced that GTPase activity caused greater tube deformation. **d** To rule out that compression was biased by the deflation state, we plotted tube diameter vs. mean pitch for FtsZ-YFP-mts ($N = 12$) (green) and FtsZ-YFP-mts*[T108A] ($N = 10$) (gray) in steady state. Despite of higher tube densities (arbitrary units) for FtsZ-YFP-mts*[T108A] as shown in (**d**—insert), the mean pitch for no GTPase case is longer at comparable tube diameters. **e** We observed two clear pitch states for FtsZ-YFP-mts (light green bars/green line) and FtsZ-YFP-mts*[T108A] (gray bars/magenta line) with a clear dominance of longer pitch for the mutant without GTPase activity. **f** Helical deformations can be understood by twisting an elastic rod subjected to constant force. We postulate that FtsZ has an intrinsic torsion that is enhanced by GTPase activity, driving further compression. Intrinsic FtsZ torsion rules long-pitch transformations ($\lambda > 3\,\mu m$) while GTP enhances further torsion causing higher pitch states ($\lambda < 2\,\mu m$).

also the amount of protein (Fig. 3d—insert). Although there was a mild correlation between pitch and diameter (Fig. 3d) for FtsZ-YFP-mts ($N = 12$), the mean pitch was consistently longer for ($N = 10$) FtsZ-YFP-mts*[T108A] (Fig. 3d) in the case of tubes with comparable or higher protein density (Fig. 3c—insert). To better visualize the impact of GTPase activity, we plotted the pitch distribution for both proteins: the GTPase activity contributed to a decrease of pitch (Fig. 3e) as clear indicative of spring compression. Interestingly, both distributions can be reasonably considered to be bimodal. We suggest that this might indicate two states of torsion: a structural intrinsic torsion (longer pitch) that is further enhanced (shorter pitch) via GTPase activity (Fig. 3f). Note that FtsZ-YFP-mts*[T108A] exhibits residual GTPase activity, driving some compression and potentially explaining the bimodality of this distribution.

To assess mechanical properties of FtsZ-YFP-mts-induced spring-like structures, we implemented an alternative approach based on the elastic response of the GUV + tube system to a specific dynamic input. Using a piezoelectric stage, we induced a lateral oscillation of the GUV position ($A = 3\,\mu m$, $f = 1\,Hz$) and recorded forces by the optical trap (Supplementary Movie S5). We here measured the resistive force of the material per micrometer (k-spring constant). The stiffer the material, the higher force detected by the optical trap. To calculate the amplitude of the signal at 1 Hz, the signal was Fast Fourier-transformed, as depicted in Fig. 4b, where the magenta line refers to the pure lipid tube and the green line to FtsZ. The pure lipid contribution ($N = 11$) yielded values between 0.15 and 0.59 pN/μm (Fig. 4d), while values between 0.23 and 1.52 pN/μm (Fig. 4d) were obtained for FtsZ. Although forces were recorded for tubes fully covered with FtsZ (Fig. 4a); for some vesicles, the lipid

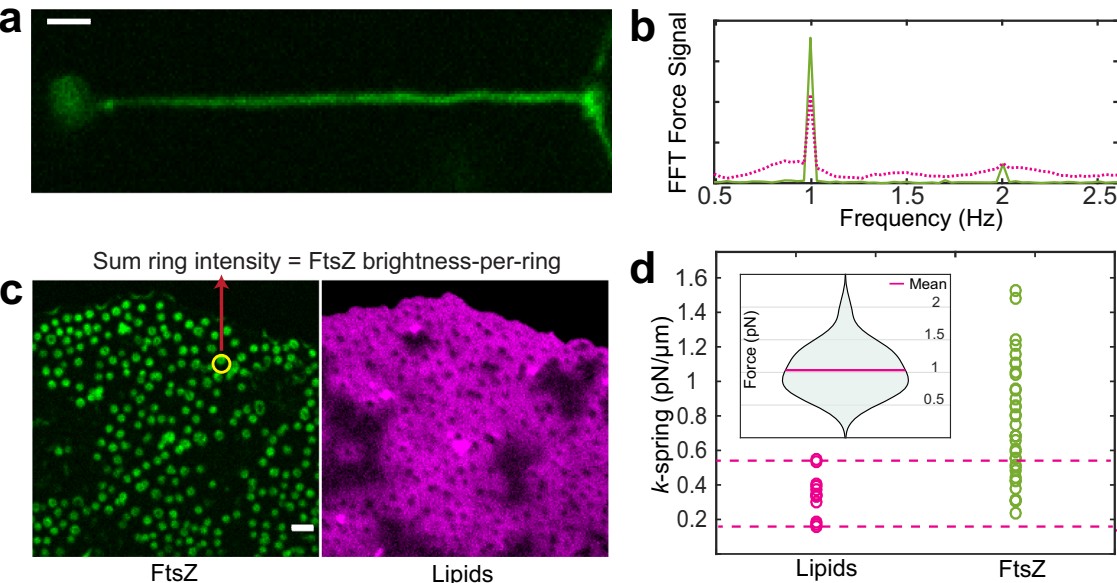

**Fig. 4 Dynamic FtsZ-YFP-mts rings exert forces in the pN range. a** Spring-like structures ($N = 36$) were mechanically assessed by forcing the tube length to oscillate with an amplitude of 3 μm and a frequency of 1 Hz. To measure a reliable force contribution from the protein, we increased the protein sample concentration (see "Methods") to guarantee a full/high protein coverage of the tube. **b** To measure forces, we tracked bead-displacement as response of the dynamic input. Then we Fast Fourier-transformed (FFT) the data to calculate the amplitude of the signal. Magenta line: lipid signal and green line: FtsZ. **c** Spontaneous flattening of vesicles over the glass surface permitted to characterize the total brightness of single rings ($N = 412$) with exactly same conditions as lipid tube experiments. **d** By calculating the amplitude of each FFT force signal (**b**), we assessed the spring constant for the case of the only lipid contribution ($N = 11$) and FtsZ ($N = 36$). Dashed magenta lines indicate the range where the lipid response dominated over the FtsZ contribution to the spring constant. (**d**—insert) The total FtsZ brightness for each data point in (**d**) was determined to approximate the FtsZ brightness-per-ring in accordance with Fig. S2E. Thus, the distribution of forces/ FtsZ brightness-per-ring was plotted ($N = 23$) showing a mean value around 1 pN per ring unit. (Scale bar = 2 μm).

response still dominated the spring constant measurement, meaning that some FtsZ data points lie in the lipid range (dashed magenta lines, Fig. 4d). This means that for some vesicles, the oscillation amplitude was not sufficient to fully unfold the membrane excess, and then the force induced by FtsZ range falls into the lipid contribution. For others, the membrane is completely unfolded, and in this case, we are able pick up the resistive contribution from FtsZ itself. Thus, FtsZ data points overlapping the lipid contribution were discarded for further analysis.

Interestingly, in our GUVs in vitro assays (also encapsulated rings, see Fig. 5d), we observed that discernible FtsZ-YFP-mts rings appeared to be of fairly similar size and brightness (Fig. S2E, with $N = 412$ analyzed rings). To estimate the range of forces that these ring units could exert in our particular assays, we took advantage of the fact that some ring-decorated vesicles flattened on the glass surface (Fig. 4b) allowing precise imaging of single rings on the surface using the same conditions as in the FtsZ coated tubes. Although the absolute number of monomers or filaments within the treadmilling ring units is unknown, the actuation of these ring-associated filaments shaped the lipid tube into a spring by exerting forces that are proportional to the here measured spring constant. Then, by knowing the spring constant, the oscillation amplitude, and the number of rings, we estimate that each ring unit exerts forces in the range of 1 pN (Fig. 4d—insert). In addition, one can estimate the dependence of this force on GTPase activity by using the pitch difference in Fig. 3e ($\triangle\lambda \sim 2$ μm), the average for the k-spring constant ($k_{FtsZ+lipid} \sim 0.9$ pN/μm), and the number of rings-per-μm (~1 ring). A force of ~0.9 pN suggests a significant contribution in the total ring-unit force resulting from GTPase activity. Note that these force estimations did not exclude an intrinsic lipid contribution.

The validity of our k-spring measurements can be evaluated by assuming the persistence length of FtsZ filaments. Previously, we

had inferred that FtsZ-YFP-mts rings on SLB are made of filamentous structures of ~0.39-μm length in an average[3]. Assuming a persistence length as 0.39 μm, FtsZ filaments would exhibit a flexural rigidity $K = 1.59 \times 10^{-27}$ Nm$^2$ ($K = l_p k_B T$[14]) that agrees well with previous experimental reports[15]. Based on this, we could assess the Young's modulus of FtsZ filaments: $E_{FtsZ} = 51.8$ MPa. ($K = EI$, where $I = \pi r^4/4$, the area moment of inertia, $r = 2.5$ nm[16]). On the other hand, the Young's modulus $E$ of a spring is related to the spring constant through $E = (kl_0)/S$, where $k$ denotes the spring constant, $l_0$ is the spring initial length, and $S$ the cross-section. By considering two independent hollow cylinders, one made of lipid bilayers (lipid bilayer thickness: ~5 nm) and one made of FtsZ-YFP-mts (one FtsZ monomer diameter: ~5 nm), the ratio $l_0/S$ was fairly constant in our tube experiments, and therefore the relationship $E_{FtsZ}/E_l = (k_{FtsZ}/k_l)$ is reasonably valid. To calculate $\frac{k_{FtsZ}}{k_l}$, we here considered raw averages for distributions shown in Fig. 4d and subtracted the lipid contribution in the case of FtsZ: $k_l = 0.34$ pN/μm and $k_{FtsZ} = 0.59$ pN/μm. Then, the ratio $\frac{k_{FtsZ}}{k_l} = 1.72$ showed good agreement compared to $\frac{E_{FtsZ}}{E_l} = 2.26$ assuming $E_l = 22.9$ MPa (lipids with bending $= 20 k_b T$)[17,18]. This confirmed that our force measurements corresponded well with previous flexural rigidity values for FtsZ fibers. In addition, our data provide further evidence that FtsZ filaments are softer than other cytoskeleton proteins such as microtubules ($K \sim 10^{-23}$) or actin ($K \sim 10^{-26}$)[19,20].

The helical nature of FtsZ and its torsional dynamics have been experimentally observed[21,22]; however, its relation to a potential mechanism of actively deforming membranes has not yet been clearly established. According to our observations, the helical membrane transformation caused in this study by FtsZ filaments can best be understood by assuming Darboux torque around the lipid tube. Darboux torques are tangential torques caused by a local mismatch between the plane defined by the filament curvature and

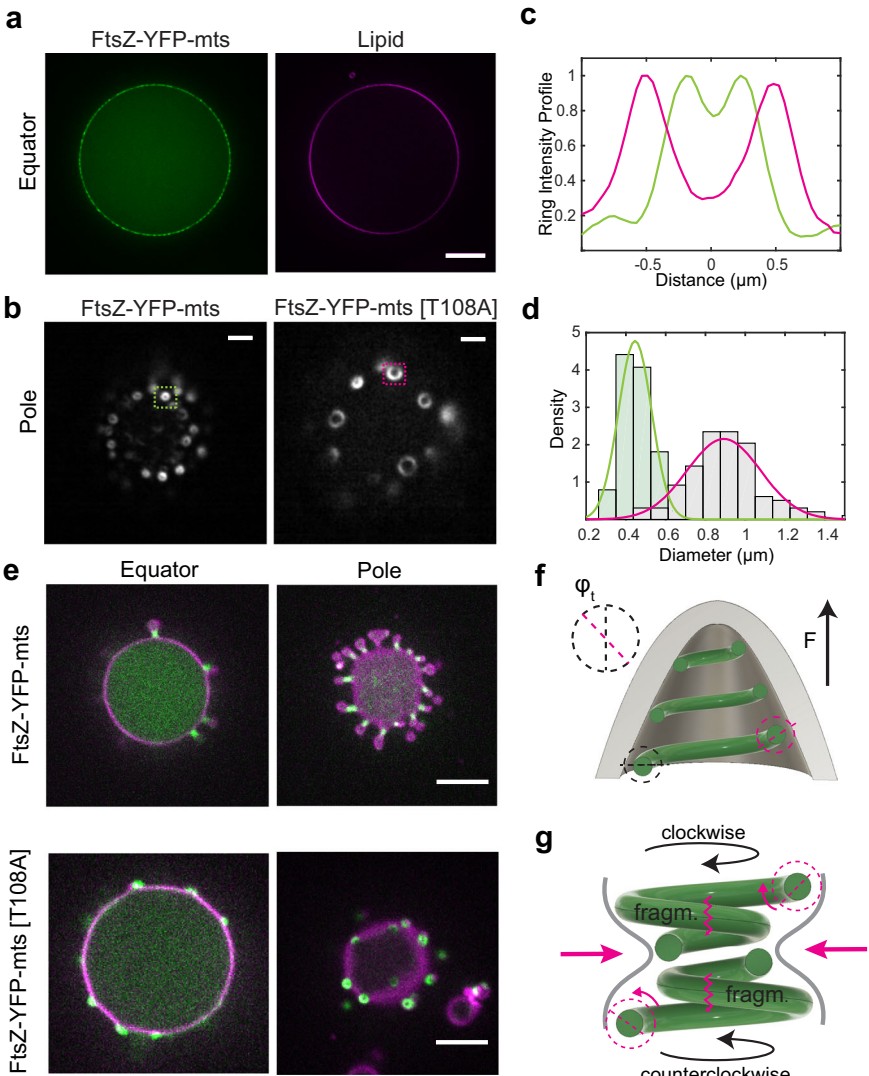

**Fig. 5 FtsZ-YFP-mts rings inside deflated GUVs cause membrane deformations that depend on GTPase activity. a** Representative image of GUVs encapsulating FtsZ (N > 10, each condition) (scale bar = 5 μm). **b** TIRF microscopy imaging of FtsZ-YFP-mts and FtsZ-YFP-mts*[T108A] rings at the bottom-inside GUVs (N > 10, each condition) (scale bar = 2 μm). **c** Intensity profile of structures indicated in **b** showed that FtsZ-YFP-mts (green line) rings exhibit smaller diameter than FtsZ-YFP-mts*[T108A] (magenta line). **d** Size distribution of (N = 112) FtsZ-YFP-mts*[T108A] (gray bars and magenta line) and (N = 102) FtsZ-YFP-mts showed a drastic reduction in ring diameter due to GTP hydrolysis. **e** After deflation, both mutants yielded outwards deformations. For the case of FtsZ-YFP-mts, N = 21/26 (80%) vesicles showed outwards deformations and constriction necks compared to N = 4/15 (26%) for FtsZ-YFP-mts*[T108A]. This suggests that GTPase activity promotes constriction and neck formation (scale bar = 5 μm). **f** We suggest that intrinsic torsion can create out-of-plane forces; however, **g** GTP hydrolysis triggered a super-constricted state favoring higher curvatures. These higher curvatures and membrane deformations cause a mechanical strain on the polymer that promotes fragmentation and then the emergence of clockwise and anticlockwise FtsZ treadmilling filaments.

the membrane attachment direction[23]. This twisting angle along the one filament is key to produce torque. A molecular dynamics study showed that dynamin, a helical endocytic constriction protein, required twisting of the "adhesive-stripe" to achieve full membrane hemifusion[23]. In the case of FtsZ, molecular dynamics studies have predicted an angle of "twisting" along the c-terminus, where membrane attachment occurs[24,25]. Also, Fierling et al. have theoretically studied membrane deformations produced by filaments inducing torques[26]. Strikingly, they found inward vortex-like deformations from flat surfaces and spring-like shapes when filaments wrapped around a tubular geometry[26]. These predictions agree remarkably well with our observations.

So far, we had investigated an inverse geometry, i.e., FtsZ added from the outside, as compared to the physiological case. Now, we also reconstituted FtsZ-YFP-mts and FtsZ-YFP-mts*[T108A]

inside GUVs (Fig. 5a). Conditions to obtain ring-like structures (Fig. 5b) or filaments wrapping the vesicle (Fig. S1D) were again found by tuning GTP and $Mg^{+2}$. Interestingly, the diameters of FtsZ-YFP-mts*[T108A] rings were significantly larger (0.89 μm) than FtsZ-YFP-mts (0.44 μm) (Fig. 5d). This difference was not observed in the case of SLBs[3], suggesting that enclosure and deformability of the lipid surface affect the steady state of FtsZ assembly. In addition, the wide size distribution in the low GTPase mutant case (Fig. 5d) might indicate that polymers were more flexible to accommodate a larger variety of curvatures. Strikingly, both FtsZ mutants could create outwards deformations emerging from rings (Fig. 5e). But only in the case of FtsZ-YFP-mts, there was clear evidence of ring constriction (Fig. 5e) in agreement with previous reports[1]. Indeed, 80% (N = 21/26) vesicles showed outwards deformations and constriction

necks simultaneously; in contrast to 26% for FtsZ-YFP-mts* [T108A].

In view of this data, we suggest that FtsZ ring formation creates outwards out-of-plane forces on deflated GUVs due to filament structure and polarity (Fig. 5f). However, FtsZ filaments only exhibiting structural torsion (low GTPase activity) are unable to stabilize smaller diameters in agreement with long-pitch spring deformations in lipid tubes. In contrast, upon GTP hydrolysis, FtsZ filaments drive constriction and condensation inside deflated GUVs in a similar way to how the same GTPase activity drives spring compression in a tubular geometry when the protein is externally added. This establishes an interesting similarity between FtsZ and dynamin, a motor protein well-known for torsional and contractile features in which GTP hydrolysis triggers a super-constricted state leading to fragmentation and clustering[27,28]. This similarity let us to hypothesize an alternative explanation for the emergence of FtsZ treadmilling and indicate a possible connection to force generation. While the leading-edge of the filament grows, a significant increase in the torsional stress over the trailing edge of the filament (due to GTP hydrolysis) drastically deforms the membrane, which in turn imposes a mechanical strain on the FtsZ filament. This strain would be highest in the membrane region between two filaments growing against each other (Fig. 5g), promoting fragmentation and bidirectional treadmilling. This also suggests that the physical properties of the membrane (e.g., membrane tension) plays a role in the emergence of FtsZ dynamic-treadmilling patterns.

Although it has been pointed out that these mechanistic studies on FtsZ in controlled membrane environments may not easily be transferred to the situation in vivo, it is tempting to evaluate the ability of FtsZ-YFP-mts construct to deform and actuate lipid membranes in a more physiological setting. Therefore, E. coli cells were transformed with a plasmid containing the corresponding gene under control of an inducible promoter. Upon IPTG induction, FtsZ-YFP-mts fluorescence signals in the cells were observed. The FtsZ-YFP-mts construct localizes in several ring-like structures around midcell (Fig. 6a). Multiple Z-ring structures were observed, due to the overexpression of the FtsZ-YFP-mts protein. A 3D-reconstruction reveals that these FtsZ assemblies are indeed ring structures that resemble those formed by native FtsZ rings at the division site (Fig. 6b). Importantly, without addition of inductor, no FtsZ-YFP-mts structures were observed (Fig. 6a). Since FtsZ-driven membrane deformations could not be observed in tensed GUVs (Fig. 5), we reasoned that they were even less likely to appear in walled bacteria with turgor pressure. Therefore, cells were treated with lysozyme to create E. coli spheroplasts in osmoprotective media. Cells expressing the FtsZ fusion protein were highly fragile and prone to lysis. We therefore started microscopic analyses before all cells have converted to spheroplasts (Fig. 6a). Importantly, vesicular structures budding out from spheroplasted cells were observed (Fig. 6a, arrows). These vesicular structures were not observed in control cells lacking the FtsZ-YFP-mts expression, indicating that they are a consequence of protein overproduction. However, in the MBL medium, the occurrence of FtsZ-YFP-mts budding was low (Fig. 6a). In contrast, similar lysozyme treatment in sucrose-buffer showed cells with (i) small outwards deformations (irregular surface) that correlate with a regions higher protein density and (ii) swelled cells with clear vesiculation (Fig. 6c and Supplementary Movie S6). A membrane stain confirmed that areas with strong FtsZ fusion protein assemblies displayed lipid membrane budding (Fig. S3) and constriction necks. Although the presence of cells with irregular surfaces (small outwards deformations), single and multiple vesiculation was not exclusive of overexpression, the frequency of such events clearly increased due to FtsZ-YFP-mts (Fig. 6d). Further interesting phenotypes

such as cells connected by lipid tubes or "pearl-necklaces" were observed (Fig. S3). These results agreed remarkably well with our outwards deformations and constriction necks from FtsZ rings inside GUVs. Directional screw-like forces promoting extrusion of lipid material or budding (Figs. 5h and 6b), as well as constriction necks (Figs. 5g and 6c), are both explained in terms of a FtsZ polymer able to exert torsional stress as explained above. Interestingly, this demonstrates the possibility of FtsZ filaments playing an active role in cell division organisms that divide by budding, such as Acholeplasma laidlawii[29].

In this work, we show that FtsZ filaments can deform lipid membranes via torsional stress due to an intrinsic twist along the FtsZ filament. In addition, we demonstrate that the GTPase activity, which is otherwise responsible for filament treadmilling, enhances this torsional stress promoting further constriction/ condensation. As a result of this interaction, FtsZ-YFP-mts rings exert forces in the range of 1 pN upon GTP hydrolysis; directional forces that suffice membrane deformations and constriction/ condensation in deflated GUVs as well as wall-less E. coli cells. Although this force range might not suffice for the entire process of bacterial cytokinesis of walled rod-like cells, given the temporal relevance of FtsZ dynamics in the coordination of synthesis of new wall material[5,6], an initial inwards membrane deformation may be key to trigger cytokinesis in the form of a FtsZ-curvature-trigger. Interestingly, viable FtsZ-GTPase mutants[6] as well as temperature-sensitive mutants[30] exhibit abnormal septum formation or twisted septum. In agreement with our data, GTPase FtsZ-deficient mutants could generate inwards membrane deformations along a helical FtsZ filament. Due to the lack of further twisting/condensation in FtsZ, the synthesis of new wall material would follow a relaxed-helix pattern (twisted septum) rather than a compressed-helix or "ring". We also hypothesize that if the membrane tension is lowered by incorporation of de novo synthesized lipids[10], the here reported forces range might become relevant for the initiation of cell division.

## Methods

**Plasmids and primers**. Please refer to Supplementary Information (Tables S1 and S2).

**Protein purification**. FtsZ-YPF-mts*[T108A] mutation was constructed using site-directed mutagenesis. T108A_RV and T108A_FW oligonucleotides were designed using NEBaseChanger–Substitution to replace the Thr in position 108 by an Ala, as described in our previous work[3]. Briefly, FtsZ-YFP-mts was first amplified using the FW and RV oligonucleotides in different PCR reactions, testing three different temperatures: 54 °C, 58.5 °C, and 65 °C. In a second PCR reaction, the PCR products from the FW and RV oligonucleotides were mixed; also, three different temperatures were tested: 54 °C, 58.5 °C, and 65 °C. After digestion with DpnI, the three PCR products were used to transform CH3-Blue competent cells. Efficient colonies were picked and confirmed by sequencing.

FtsZ-YFP-mts and FtsZ-YPF-mts*[T108A] were purified as described[3]. Briefly, the protein was expressed from a pET-11b expression vector and transformed into E. coli strain BL21. Overexpression was performed at 20 °C for the proteins FtsZ-YFP-mts. Cells were lysed by sonication and separated by centrifugation. Then, protein was precipitated from the supernatant, adding 30% ammonium sulfate and incubating the mixture for 20 min on ice (slow shaking). After centrifugation and resuspension of the pellet, the protein was purified by anion exchange chromatography using a 5 × 5-ml Hi-Trap Q-Sepharose column (GE Healthcare, 17515601). Protein purity was confirmed by SDS-PAGE and mass spectrometry.

Wild-type FtsZ was purified from E. coli cells (BL21 strain) using the plasmid pET-28a containing the E. coli FtsZ gene as previously described[31]. Briefly, overexpression of wild-type FtsZ was performed at 37 °C for 3 h after induction with IPTG. Cells were lysed, and the soluble fraction was separated by centrifugation. The protein was extracted by two cycles of $Ca^{2+}$-induced precipitation. First, the sample was incubated during 15 min at 30 °C after addition of 1-mM GTP and 20-mM $CaCl_2$. The sample was centrifuged and resuspended, followed by a second cycle of $Ca^{2+}$-induced precipitation. The protein was then purified by anion exchange chromatography using a 5 × 5-ml Hi-Trap Q-Sepharose column (GE Healthcare, 17515601).

The soluble six-His tags (sZipA) construct of ZipA was produced by elimination of the hydrophobic N-terminal domain (first 25 amino acids) as described previously[32]. Having the plasmid pET-15ZIP as template DNA, deletion of amino

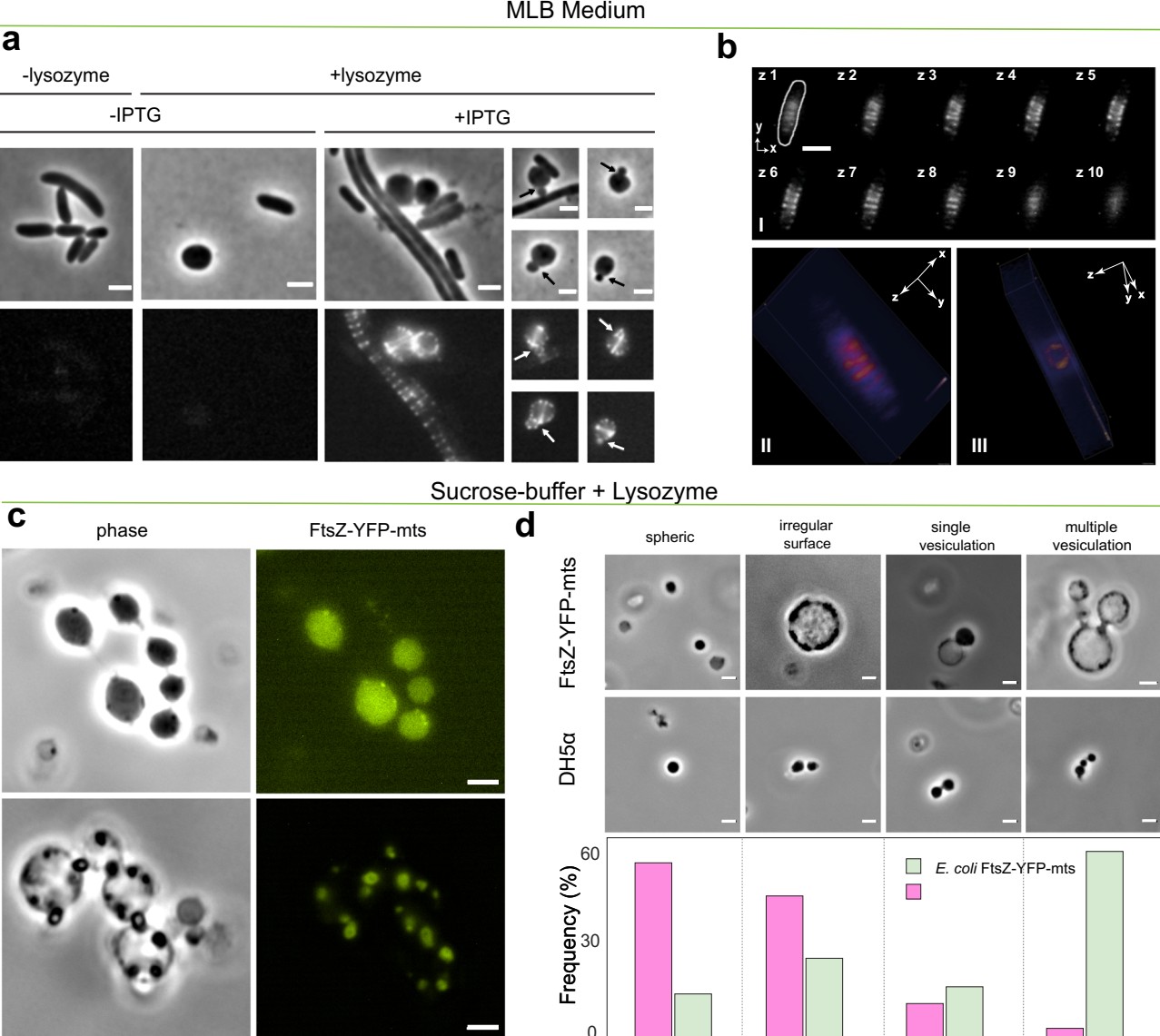

**Fig. 6 In wall-less *E. coli* cells, FtsZ-YFP-mts rings exert outward forces causing budding, vesiculation, and constriction necks. a** *E. coli* DH5α cells expressing FtsZ-YFP-mts show impaired division and a regular fluorescence pattern. Removal of the cell wall by lysozyme treatment leads to spheroplast formation. Occasional membrane vesiculation can be correlated to the localization of FtsZ-YFP-mts (arrows) (scale bar = 2 μm). **b** Upon induction, cells express FtsZ-YFP-mts polymeric structures perpendicular to the cell length, around midcell (bI). 3D rendering reveals ring-like structures (bII–bIII) (scale bar = 2 μm). **c** Attempts to enhance the spheroplasting efficiency (sucrose-buffer) resulted in the emergence of multi vesiculated structures. Points of membrane constriction correlate with presence of FtsZ-YFP-mts. Deformations of the plasma membrane indicate a force generation by FtsZ assemblies that leads to local membrane invaginations and eventually pinching off of vesicles (scale bar = 2 μm). **d** Quantitative morphological classification of induced *E. coli* DH5α *pEKEx2-ftsz-yfp-mts* cells (*N* = 50 cells) compared to *E. coli* DH5α (*N* = 43 cells), after lysozyme treatment in sucrose-buffer, within the four categories: spheric, irregular surface, single and multiple vesiculation. (Scale bar = 2 μm).

acids was obtained by inverse polymerase chain reaction (PCR) using primers sZipAI and ZipAII. After PCR purification, the product was digested with DpnI, and ligated with T4 DNA ligase. Overexpression of sZipA was induced with IPTG for 3 h at 37 °C in transformed BL21 cells carrying the pET-15ZIP cloning vector. After cell lysis and centrifugation, the soluble fraction was loaded on a 5-mL Ni-NTA resin column (Novagen) and the protein was eluted in buffer containing imidazole. Protein purity was confirmed by SDS-PAGE and mass spectrometry.

Wild-type FtsZ and sZipA were covalently labeled in the amino groups with Alexa Fluor 488 carboxylic acid succinimidyl ester dye (Molecular Probes/Invitrogen) as earlier stated[33].

**Imaging chamber for slow evaporation.** The imaging chamber (Fig. S4A) was made using—customized size—coverslip glasses (Menzel, Germany) and a PMMA 3-mm height spacer (Fig. 1). The bottom coverslip has 1.5 thickness and 12-mm width. The upper one was 1.0 thickness and 10-mm width. Glued coverslips to the spacer form a two-side-open chamber of ~380-µl volume. Coverslips were cleaned

by immersion in a 50:50 water/ethanol solution followed by 1-min air-plasma cleaning. After this, coverslips were glued to the spacers and passivated for more than 2 h using 5-mg/ml b-casein (Sigma, Germany). Samples are examined for 1 h with a consistent evaporation osmolarity ~50 mOsm per hour under constant humidity and temperature conditions. Examined vesicles were located in the central area of the chamber, fairly distant from the evaporation meniscus.

**Confocal spinning disk imaging and optical tweezers setup.** Confocal spinning disk imaging was performed using a Yokogawa CSU10-X1 spinning disk system connected to a Nikon Eclipse Ti inverted microscope (Nikon, Japan). A DPSS laser stack with 488, 561, and 640-nm laser lines (3i, Denver, Colorado USA) is used for illumination and an Andor Ixon Ultra 512 × 512 EMCCD camera for fluorescence detection. 3i-Slidebook software controlled lasers, filter selection, microscope, *xy* stage, and acquisition (3i, Denver, Colorado USA). For simultaneous FtsZ-YFP-mts and ATTO655-DOPE excitation, 488 and 640-nm lasers were rapidly switched by an AOTF. For image detection, the use of an appropriated quad-band filter avoided

mechanical filter switch. Samples were images using a CFI Plan Apochromat Lambda 100X oil, NA = 1.45 (Nikon, Japan).

The optical tweezers setup (Fig. S4B) was built according to previous reports[34,35]. Briefly, a 3W 1064-nm DPSS laser (Cobolt, Sweden) was used as trapping laser. A half-wavelength plate ($\lambda/2$ plate) and a polarizing beam splitter control the laser power entry to the scope. A two-lens telescope expands the laser beam to fill the back aperture of—above mentioned—100× objective. In addition, two mirrors adjust the laser-beam $xy$ position. The force detection arm was mounted in the illumination pillar of the microscope, collecting (from above) the light from the laser using a—long working distance—condenser 20X, NA = 0.6 (Edmund optics, USA). A micrometric $xyz$ stage positions the condenser (and the detection arm) in -z- to achieve Köhler illumination and -xy- for centering. For alignment, once the proper laser axis has been defined, the position of field diaphragm (having Köhler illumination) is imaged through an auxiliary camera DCC3240M (Thorlabs, USA) on the right port of the microscope. Using camera software options, a drawing masking the field diaphragm was saved in a file. This drawing can be reused in such a way that the drawing overlays the real-time camera acquisition. Therefore, $xyz$ knobs are moved to position the field diaphragm according to the drawing mask.

In the detection arm, a relay lens imaged the back focal plane of the condenser in a quadrant positional detector (QPD) PDQ80A (Thorlabs, USA). The signal voltage of the QPD defined the position of the trapped bead. The force detection module OTKBFM (Thorlabs, USA) was used for data acquisition. This DAQ acquired a voltage signal from the QPD at a maximum rate of 100 kHz. In addition, this module was customized to control a $20 \times 20$-µm range LT2 piezoelectric stage (Piezoconcept, France) using an analog signal (1–10 V). Position calibration and trap-stiffness determination was performed using the OTKBFM software (Thorlabs, USA). To calibrate position, beads were attached to a clean glass coverslip in high salt medium (1-M NaCl). Here, the streptavidin-coated beads have a mean diameter of 1.71 µm (Spherotech, USA). The laser beam is focused over one bead. The $xy$ position of the bead, relative to laser beam, was adjusted using the piezo stage to maximize the signal in QPD. Similarly, the focus -z- position of the laser beam was adjusted to maximize QPD output. Then, the piezo stage was moved $4 \times 4$-µm $xy$ and changes in QPD signal were recorded. From the typical "s-shape" response curve, a straight line was fitted representing the volts to µm conversion. The trap stiffness was determined using the power-spectrum analysis. Using this, the frequency corner of a $xy$ Brownian motion subjected to a trapping potential was fitted from data. The stiffness of the trap was measured over different laser powers to determine the linear range. All experiments are performed at the same laser power of 200 mW (before objective) with a 1.66-V/µm conversion and a trap stiffness of $74.4 \pm 2.5$ pN/µm. Experiment acquisition was performed using a Matlab script controlling the OTKBFM. This Matlab script set (for 60 s) an oscillatory motion for the piezo stage with an amplitude of 3 µm and a frequency of 1 Hz. In parallel, QPD data were acquired at a rate of 10 kHz.

### FtsZ rings reconstitution in vesicles

*FtsZ externally added to vesicles and tube experiments with force detection.* GUVs were produced using electro-swelling. *E. coli* lipid extract was dissolved in chloroform to reach a concentration of 3 mg/ml. To image lipids in the red channel, 0.05% (mol) of ATTO655-DOPE (ATTO-Tech GmbH, Germany) was added to the lipid mixture. In addition, 0.15% of DSPE-PEG (2000) Biotin (Avanti, AL, United States) was also added to achieve binding between GUVs and streptavidin-coated beads. For experiments with Wt-FtsZ, 0.5% DGS-NTA lipids were also added to enable the attachment of sZipA protein. In our home-made Teflon chambers, three drops (~1 µl) were carefully seeded in Pt wires and rapidly air-dried. After 1-h vacuum of further chloroform drying, GUVs were swelled in 250-mOsm sucrose solution at 10 Hz for 2 h and 2 Hz for 1 additional hour (detachment). Lipid concentration and electroformation times were carefully optimized to guarantee an appropriated GUV yield. Furthermore, to obtain similar GUV-lipid concentrations, two identical chambers underwent same procedure and mixed afterwards.

GUVs were mixed in buffer (120-mM KCl, 20-mM Tris-HCl, and 1.5-mM MgCl$_2$ pH 7.5) with an osmolarity of 250 mOsm. In details, 4 µl of GUVs were highly diluted in 360 µl of buffer in a reaction tube. FtsZ-YFP-mts and FtsZ-YPF-mts*[T108A] were added to reach a final concentration around 0.07 µM. For experiments with Wt-FtsZ and siZipA, the same procedure was used adding first siZipA at around 0.2 µM followed by addition of FtsZ reaching ~0.8 µM supplemented with ~0.2-µM wt-FtsZ-Alexa 488. In both cases, polymerization is triggered by adding 1.25 mM of GTP (Sigma, Germany). Flaccid vesicles with FtsZ rings and inward deformation were observed after 20 min. On these vesicles, tubes were pulled and imaged for 10 min (1 fps) in two colors: FtsZ-YFP-mts/wt-FtsZ-Alexa 488 and lipid channel. This time was enough to obtain helical deformations and reach a steady state of protein entering the tube, at least for FtsZ-YFP-mts. Right after, force measurements (oscillations) were performed for 60 s. Experiments with pulled tubes longer than 10 min were prone to display contamination on the "force channel" (beads, small vesicles, and protein clusters were attracted to the trapping bead). This made force measurements with FtsZ-YFP-mts*[T108A] difficult since the timescale of deformation was significantly longer. To analyze pitch, $N = 12$ experiments for FtsZ-YFP-mts and $N = 10$ for FtsZ-YFP-mts*[T108A] were analyzed.

To measure forces and estimating spring constants, we used the oscillation mode as described above. Force experiments were carried out in tubes without protein to be contrasted to experiments with FtsZ-YFP-mts and helical deformations. As mentioned above, data were acquired in Matlab (Mathworks, USA) and signal amplitude determined by the amplitude of the FFT (Fast Fourier transformation) at 1 Hz. Then, these values are converted to force using volts/µm conversion and the trap stiffness and further divided by the oscillation amplitude ($A = 3$ µm). To guarantee high/full protein coverage of the tube in FtsZ force experiments, we increased the protein sample concentration to 0.08–0.09 µM. To characterize single FtsZ ring brightness, we imaged vesicle (outside FtsZ) rings that have flattened over the glass surface. Using same light conditions as used for tube experiments, the brightness of FtsZ rings of $N = 421$ rings was analyzed and plotted (Fig. S2E).

### FtsZ encapsulated in lipid vesicles

FtsZ encapsulating vesicles were produced using droplet emulsion transfer[36]. Lipid composition was EggPC/DOPG 80:20 (Avanti, AL, United States) with 0.05% mol ATTO655-DOPE (ATTO-Tech GmbH, Germany). Briefly, vacuum-dried lipids were dissolved in mineral oil (Sigma, Germany) to reach a final concentration of 0.5 mg/ml. To form lipid vesicles, two interfaces are required: outer and inner interface. In a reaction tube (A), 500 µl of lipid + oil mixture was added to 500 µL of outer buffer (150 KCl 50-mM Tris-HCl pH 7.5). At the oil–water interface, a lipid monolayer was assembled after 30 min. In a second reaction tube, 15 µl of protein master mix was added 500 µl of lipid + oil and vigorously vortexed for 2 min to obtain a homogenous cloudy emulsion. The inner monolayer was rapidly formed (~2 min). This protein master mix was composed of inner buffer (125-mM KCl, 25-mM Tris-HCl, 2-mM MgCl$_2$ pH 7.5), 20% OptiPrep (Density Gradient Medium, Sigma, Germany), protein and GTP. FtsZ-YFP-mts (or FtsZ-YPF-mts*[T108A]) and GTP final concentrations were 1.65 µM and 1.4 mM, respectively. Therefore, the emulsion was transferred to the reaction tube (A) and centrifuged at 100 $g$ for 7 min. Finally, the oil-based supernatant is discarded and 300-µl final vesicles are 1:2 or 1:3 diluted in fresh outer buffer.

### Image analysis

Image analysis and plotting were carried out in MATLAB (MathWorks, USA) and ImageJ (NIH, USA).

To generate kymographs (Fig. 1), a script allows the user to define a circular section by providing two coordinates such as described in reference[3]. This circular section is automatically fitted to a circle with radius $r$ (in this case, $r = 0.5$ µm). Then, three trajectories corresponding to three concentric circles having radii $r$, $r + 1$, and $r - 1$ pixels are determined. At this point, the script read the time-series data and calculate a kymograph for each time point and trajectory. The final kymograph corresponds to the average of the three different trajectories[3].

To determine tube diameter (Fig. S2A), the intensity over a representative tube section was normalized to be fitted to a Gaussian function. Then, the here reported diameter represented half-width of the Gaussian. To quantify the arclength (helical shapes projection to an 2D image), tubes were binarized and fitted to a function using linear interpolation (Fig. 3). Then, the arclength of the function was calculated. The protein density was calculated by measuring, over the tube, the FtsZ-intensity difference between initial and final acquisition. This difference was divided by the FtsZ intensity on the GUV (in the tube proximity) times the tube length.

To measure the pitch in the spring-like tubes (Fig. 3), a Gaussian filter was applied. The tube red-lipid-intensity profile was integrated and normalized, in a perpendicular direction (to the tube), to find the pixel position where the normalized intensity was "1" and closer to "0.3". The 1 (maximum) position pixel defined the center of the tube while "0.3" defined the upper and lower limit of the tube. The intensity profile, now in parallel direction to the tube, was plotted for these three locations: up, center, and down. Using this intensity profile, peaks are automatically found and then pitch was calculated. Every experiment is manually inspected to avoid miscalculation.

To measure the size distribution of FtsZ encapsulated rings (Fig. 5), TIRF imaging was used according to our previous work[3]. The diameter was manually measured using intensity profile in ImageJ (NIH, USA). Diameters were exported and plotted in MATLAB. For both protein mutants, the number of analyzed rings was $N > 100$.

### In vivo assays, spheroplasts generation, and imaging

For overexpression of FtsZ-YFP-mts in *E. coli* cells, the respective DNA sequence was amplified by PCR using oligonucleotides SacI-mts and SalI-ftsZ, respectively. The pET-11b-ftsz-yfp-mts plasmid served as template DNA. The resulting PCR product was ligated into SacI/SalI opened pEKEx2-vector. The final plasmid pEKEx2-ftsZ-YFP-mts was transformed into competent *E. coli* DH5α cells.

*E. coli* cells expressing FtsZ-YFP-mts were cultured in osmoprotective MLB-medium (1% peptone, 0.5% yeast extract, 1-mM CaCl$_2$, 30-mM glucose, 25-mM MOPS (pH 7.2), 340-mM NaCl)[37]. Plasmid maintenance was ensured by addition of 25-µg/ml kanamycin and cultures were incubated at 37 °C with constant shaking (120 rpm). Expression was induced by adding 100-µM IPTG. The formation of sphaeroplasts was induced by addition of 0.5-mg/ml lysozyme, followed by an incubation of 30 min at 37 °C. For microscopy, a 0.1% agarose-pad in MLB medium was used.

In order to improve spheroplasting efficiency, an adaptation of the method described in ref. [38] was used. Therefore, single colonies were picked and cultivated

overnight in 10-ml LB-medium (10-g/l tryptone, 5-g/l yeast extract, and 10-g/l NaCl), using environmental conditions like described above. The next day 200 µl of the overnight culture were used to inoculate the respective over day culture. For the mutant, kanamycin (25 µg/ml) was used in both pre-cultures, induction with IPTG was performed on the over day culture only. After 4 h, 1 ml of exponentially growing cells was harvested at 3000 g for 1 min. The pellet was gently resuspended in 0.8-M sucrose, together with 30-µl Tris-HCl (pH 8.0), 24-µl 0.5-mg/ml lysozyme, 6-µl 5-mg/ml DNaseA and 6-µl EDTA-NaOH (pH 8.0) and incubated for 5 min at room temperature. One hundred microliter of a STOP solution (10-mM Tris-HCl (pH 8.0), 0.7-M sucrose, and 20-mM MgCl₂) were added to complex-free EDTA. For one replicate, phospholipid membranes were stained for 5 min with 1-µg/ml nile red (Invitrogen). For microscopy, 2 µl of the suspension were directly applied on the sample slide with a cropped pipet-tip and covered with a high-precision coverslip.

Fluorescence and phase contrast imaging for in vivo studies were performed on a Zeiss Axio Imager M1 microscope equipped with an immersion condenser, a 2,5x optovar and an EC Plan Neofluar 100x/1.3 Oil Ph3 objective (Zeiss). YFP fluorescence was imaged using filterset 46 HE shift free (EX BP 500/25, BS FT 515, EM BP 535/30), and nile red fluorescence was detected with filter 43 HE Cy 3 shift free (EX BP 550/25, BS FT 570, EM BP 605/70). Image acquisition was carried out using the AxioVision software-package (Zeiss). For image analysis, FIJI was used[39].

**Reporting summary**. Further information on research design is available in the Nature Research Reporting Summary linked to this article.

## Data availability
All other data and materials supporting this publication are available from the corresponding author upon reasonable request. Source data are provided with this paper.

## Code availability
Custom code has been deposited in https://github.com/diegoalejandrord/FtsZ_Torsion_NatComm (https://doi.org/10.5281/zenodo.4695237)[40].

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

## Acknowledgements
We acknowledge Sven Vogel, Allen Liu, and Ethan Garner and for very useful discussions in addition to Rudy Mendez and Sophie Dvali for experimental help. We thank to Daniela Garcia, Ana Raso, and the MPIB Core Facility for assistance in FtsZ-YFP-mts protein purification. This work was funded through MaxSynBio (MPG together with Federal Ministry of Education and Research of Germany) Grant Number 031A359A to P.S., and the Transregio CRC 174 by the DFG (Deutsche Forschungsgemeinschaft) to P.S. and M.B. Work at GR Lab was supported by Spanish Government Grants BFU2016-75471-C2-1-P and 2019AEP088.

## Author contributions
D.A.R.-D. and P.S. conceived this study and wrote the manuscript with the support of M.B. for the in vivo spheroplasts section. D.A.R.-D. designed, performed, and analyzed all experiments with FtsZ-YFP-mts and FtsZ-YFP-mts*[T108A]. A.M.-S. designed, performed, and analyzed experiments with ZipA + FtsZ using proteins provided by G.R.

M.H. manufactured wafer for microstructures and provided PMDS microstructures. M.B. and F.M. designed, performed, and analyzed in vivo spheroplasts experiments. MATLAB custom scripts for data acquisition and analysis were coded by D.A.R.-D. All authors discussed and interpreted results and revised the manuscript.

## Funding

## Competing interests

The authors declare no competing interests.
