## [Peer Review File · Nature Communications]

Reviewers' Comments:

Reviewer #2:

Remarks to the Author:

The revised manuscript has greatly improved their statistic power by adding sample sizes and percentages in their initially descriptive figures. It also improved clarity of the writing. However, I am still very reluctant to agree with the authors that the further deformation of the lipid tubes is indeed caused by the treadmilling dynamics of FtsZ.

The major evidence the authors used to support this conclusion is the smaller helical pitches that formed at the later stage of pulled lipid tubes, i.e. FtsZ probably first polymerized on the tubes to form helical structures, and this helical structures further compressed due to its treadmilling dynamics. This observation is in contrast to the GTPase mutant, which only formed helical structures with larger pitches and without further compression.

These observations are certainly consistent with the authors' hypothesis, but do not prove it—they are also consistent with the possibility that the difference between the two pitches of the WT and mutant FtsZ is caused by the intrinsic polymer structure but not the treadmilling dynamics.

I suggested in the last review that the authors should conduct a control experiment in which they use WT FtsZ with nonhydrolyzable GTP to test this idea, but the authors argued that they could not find a condition to form rings on supported lipid bilayers. Did they try it in their deflated liposomes instead?

If the experiment with nonhydrolyzable GTP is truly difficult, the authors should make good use of their ZipA-FtsZ construct; FtsZ filaments formed with ZipA as the membrane anchor does not exhibit the same treadmilling dynamics as the one with FtsA (or the mts anchor the authors used, Loose, Nat Cell Biol, 2013), therefore this experiment could serve as a pseudo control for FtsZ filaments that

still have WT GTPase activity but is somehow diminished in treadmilling activity (this can also be tuned by using higher ZipA concentrations). Unfortunately, the authors only said that the experiment with ZipA showed similar helical structure as mts, but there was no quantitative description of whether there is a further compression phase, or whether the helical pitches are indeed the same. The fact the authors observed plectonic/supercoiled regions of the tubes probably suggests that the treadmilling dynamics may not be absolutely necessary for this further compression.

The authors also did not really address the issue of converting fluorescence to concentrations. There was a description of "Although the absolute number of monomers or filaments comprising these treadmilling ring units is unknown, we can assign the measured forces of around 1 pN driving the transformation of a lipid tube into a helical structure to these discernible FtsZ-YFP-mts ring units". It is not clear to me how the force could then be assigned to those lipid tubes.

I have also asked "Another control where WT FtsZ (perhaps together with FtsA/ZipA) overexpression without mts should be included to show that indeed budding is induced by FtsZ overproduction... ". This point was not addressed in the response. This should be a relatively straightforward and critical experiment. Probably they should also do the overexpression experiment with the GTPase mutant to support their claims that only the treadmilling ability causes these bubbling effect.

Some other minor points:

1. In their response: "We did not observe any further changes after 5 minutes. In contrast, a steady state for the T108A mutant was reached between 10-20 mins. ". I think this is important for the readers and they should have included it in the main text.

2. In their main text Line 125, they need to cite their previous work there.

Reviewer #3:

Remarks to the Author:

The authors have significantly improved and clarified the manuscript. However, a few aspects remain rather confusing:

- The authors now state that spring constants have been converted to forces by multiplying by the amplitude of the oscillation. This means that the obtained value will depend on the applied oscillation, and thereby rises the question of why the specific amplitude of 3 μm was chosen. Is it because it fully unfolds the membrane undulations? Also, this important clarification is mentioned in the response to reviewers but not in the text. In the text, an estimated value of 1 pN is now mentioned, without any clear explanation.

- Regarding cross sections of the tubes, I thank the authors for clarifying their calculations. However, can the authors clarify how the flexural rigidity is calculated from persistence length? Also, please note that in line 184 they mistakenly state that $E = Kl$, the correct expression is $K = EI$.

- Finally, in page 7 the authors state that they discard spring constant values of FtsZ tubes that are in the range of membrane only tubes. In my view, this is not correct: by definition, this approach will lead to differences between FtsZ and lipid tubes, even if they are not there. From the data points, it is clear to me that there are indeed differences, so this is not a major concern, but I think the authors should:

- o Quantify spring constants considering all values, the fact that some FtsZ values fall within the lipid range is to be expected considering the high variability observed in both cases.

- o Then, I think it is fine for the authors to subtract both obtained spring constants to estimate the FtsZ contribution, as they do.

Reviewer #2:
Remarks to the Author:

The revised manuscript has greatly improved their statistic power by adding sample sizes and percentages in their initially descriptive figures. It also improved clarity of the writing. However, I am still very reluctant to agree with the authors that the further deformation of the lipid tubes is indeed caused by the treadmilling dynamics of FtsZ.

The major evidence the authors used to support this conclusion is the smaller helical pitches that formed at the later stage of pulled lipid tubes, i.e. FtsZ probably first polymerized on the tubes to form helical structures, and this helical structures further compressed due to its treadmilling dynamics. This observation is in contrast to the GTPase mutant, which only formed helical structures with larger pitches and without further compression.

These observations are certainly consistent with the authors' hypothesis, but do not prove it—they are also consistent with the possibility that the difference between the two pitches of the WT and mutant FtsZ is caused by the intrinsic polymer structure but not the treadmilling dynamics.

We appreciate that the reviewer acknowledges a significant improvement of our paper. Regarding the remaining skepticism, we believe there is a misunderstanding. Indeed, treadmilling is a property that is associated to GTPase activity implying a structural change in the polymer. FtsZ intrinsic polymer structure can deform membranes as we showed here; however, GTPase activity/treadmilling **enhances** this membrane transformation potency, due to a conformational change that involves chemical energy transformation.

In our previous work on supported lipid membranes (Ramirez-Diaz, D. A. et al. Treadmilling analysis reveals new insights into dynamic FtsZ ring architecture. PLoS Biol 16, e2004845 (2018).), we showed that the treadmilling is a property of the FtsZ polymer that depends on GTPase activity. This means that FtsZ rings with GTPase activity exhibit treadmilling (panel A), in contrast to just ring-forming mutants (i.e., with the right curvature to form a ring) which do not treadmill (panel C). Remarkably, there is a structural change in the polymer that leads to breakage/disassembly in the trailing edge of the polymer if, and only if, there is GTPase activity. This has to happen at a polymer level.

What could these structural changes be? Some papers have suggested that GDP forms relate to “more curved filaments” while GTP forms relate to “more straight filaments” (For example Lu, C., Reedy, M. & Erickson, H. P. Straight and Curved Conformations of FtsZ Are Regulated by GTP Hydrolysis. *J. Bacteriol.* 182, 164–170 (2000)) However, filament appearance significantly changes with buffers, crowders, etc. There is still much ambiguity that has to be solved by structural biology. On top of that, these EM images are not taken on a lipid surface, which is a big difference compared to our conditions.

Here we suggest that the polymer that forms rings but does not treadmill transform a lipid tube into a helix (T108A). And more importantly, the same polymer with GTPase activity (i.e. when treadmilling) promotes compression and condensation (we call it “extra torsion”). It is not trivial to match this observation with structural-molecular changes; however, we believe the reviewer is neglecting parts of the phenomenon. e.g., Fig. 5 also agrees with our interpretation. It is not a coincidence that the same polymers form rings (Fig 5.) with sizes that depend on GTP, i.e. with smaller diameter upon GTPase activity. This must imply that there is constriction in the plane of the ring, as well as a force perpendicular to this plane. This agrees remarkably well with the idea of GTP dependent compression/condensation

To emphasize these arguments above, we have decided to change the title of the paper and make some changes in the text.

I suggested in the last review that the authors should conduct a control experiment in which they use WT FtsZ with nonhydrolyzable GTP to test this idea, but the authors argued that they could not find a condition to form rings on supported lipid bilayers. Did they try it in their deflated liposomes instead?

In spite of the significant problems induced by the pandemic to carry out additional experiments, we addressed this point once again. Indeed, we did not find a condition to observe rings on SLBs. As suggested by the reviewer, we then performed experiments with GMPCPP on GUVs. After GMPCPP addition, we observed a rapid protein binding to the membrane that in the course of minutes resulted in the loss of ring structures and the formation of a high-density mesh of filaments, which solidified over time. In contrast, the rings persisted over of hours and remained dynamic in the presence of GTP. Because of the completely different phenomenology of the FtsZ coverage in the case of GMPCPP, attempts to pull tubes from these GUVs failed. The lower figure exemplified that, as we believe, convincingly.

This observation agrees with our SLB experiments in the way that transitions from low-intermediate density (rings) to high density (bundles, spirals) were fast. We neither found a condition of slow protein binding to the membrane and the respective possibility to resolve circular structures.

To emphasize that our experiments with the mutant T108A are addressing exactly the GTPase activity and the possibility of treadmilling, we here append a graph from the supplementary material of our earlier SLB study (Ramirez-Diaz, D. A. et al. Treadmilling analysis reveals new insights into dynamic FtsZ ring architecture. PLoS Biol 16, e2004845 (2018).). Please notice that the GTPase activity is significantly reduced; yet the protein still forms ring-like structures.

If the experiment with nonhydrolyzable GTP is truly difficult, the authors should make good use of their ZipA-FtsZ construct; FtsZ filaments formed with ZipA as the membrane anchor does not exhibit the same treadmilling dynamics as the one with FtsA (or the mts anchor the authors used, Loose, Nat Cell Biol, 2013), therefore this experiment could serve as a pseudo control for FtsZ filaments that still have WT GTPase activity but is somehow diminished in treadmilling activity (this can also be tuned by using higher ZipA concentrations).

Unfortunately, the reviewer is technically incorrect regarding the ZipA. It seems there is a general misconception in the field: one cannot decouple treadmilling and GTPase activity. Loose's early work suggested that treadmilling is diminished by ZipA or absent in FtsZ-YFP-mts. However, recent studies have shown that FtsZ +ZipA and FtsZ-YFP-mts also form treadmilling rings (similar velocities as the in vivo situation) under proper conditions. First, we showed that FtsZ-YFP-mts self-assembles into treadmilling rings as a function of protein surface density (Ramirez-Diaz, D. A. et al. Treadmilling analysis reveals new insights into dynamic FtsZ ring architecture. PLoS Biol 16, e2004845 (2018).). Another study showed that FtsZ+ZipA also assemble into dynamics treadmilling rings with a similar velocity to the in vivo situation (Krupka, M., Sobrinos-Sanguino, M., Jiménez, M., Rivas, G. & Margolin, W. Escherichia coli ZipA Organizes FtsZ Polymers into Dynamic Ring-Like Protofilament Structures. mBio. 9, (2018).).

In view of this, the suggested experiment seems not to be pertinent.

Unfortunately, the authors only said that the experiment with ZipA showed similar helical structure as mts, but there was no quantitative description of whether there is a further compression phase, or whether the helical pitches are indeed the same. The fact the authors observed plectonic/supercoiled regions of the tubes probably suggests that the treadmilling dynamics may not be absolutely necessary for this further compression.

We quote reviewer 2 in their first revision for NCB:

“It is however likely that FtsZ causes inner membrane deformation, which provides a spatial cue for cell wall enzymes to follow. In such a scenario, it is unclear whether the force measured in vitro using lipids of a specific composition and polymerization condition will be relevant in vivo. Here FtsZ is also artificially attached to the membrane using a mts, which is different from in vivo”

We performed the ZipA+FtsZ experiments to reply to that. Rather than investigating the differences the springs made by ZipA+FtsZ and FtsZ-YFP-mts, we emphasize that the torsion-phenomenology is also observed in a more “in vivo” biochemical setting. This is a yes or no question, and thus, there is no need to go further.

Lastly, the statement: *“The fact the authors observed plectonic/supercoiled regions of the tubes probably suggests that the treadmilling dynamics may not be absolutely necessary for this further compression”* is incorrect since ZipA+FtsZ has the same GTPase activity and thus, the same treadmilling phenotype, in contrast to what has been claimed earlier.

We here attach one image of treadmilling ZipA+FtsZ rings from Krupka et al (2018) and a supplementary figure from Loose and Mitchison that relates GTPase activity for different proteins.

The authors also did not really address the issue of converting fluorescence to concentrations. There was a description of “Although the absolute number of monomers or filaments comprising

these treadmilling ring units is unknown, we can assign the measured forces of around 1 pN driving the transformation of a lipid tube into a helical structure to these discernible FtsZ-YFP-mts ring units". It is not clear to me how the force could then be assigned to those lipid tubes.

The reviewer is correct to indicate that this issue has not been addressed.

We agree and understand the relevance of this point, but we strongly believe that this is out the scope of this study. One can be tempted to determine the protein-lipid binding kinetics, to estimate the concentration of protein per vesicle (given sample protein and lipid concentration) and then correlate this number to the camera signal. Precise and careful measurements are here needed. None of these assays, which require much more extensive methodological work, were established for this study, and their precise and careful implementations are matter of future research. To appreciate the challenge and the need of further methodologies, please bear in mind that the total protein concentration is 70 nM, we do not know how much of this protein is bound to liposomes. In addition, due to GTP consumption, the amount of bound protein changes over time.

In contrast, there is a clear qualitative pattern in our experiments: the brightness and size of our rings is reasonably constant. We hope the reviewers agree that by integrating the signal of the tube, we can convert this number into an estimation of the number of rings. Now, since we estimated the number of rings per tube, the filaments associated to those rings have to exert a force to shape the lipid tube into a helical spring. The energy required to shape this tube into a spring is proportional to the spring constant. Then, given the spring constant and the amplitude of the oscillation, we could estimate the force per ring.

Further clarifications will be implemented in the text.

I have also asked "Another control where WT FtsZ (perhaps together with FtsA/ZipA) overexpression without mts should be included to show that indeed budding is induced by FtsZ overproduction... ". This point was not addressed in the response. This should be a relatively straightforward and critical experiment. Probably they should also do the overexpression experiment with the GTPase mutant to support their claims that only the treadmilling ability causes these bubbling effect.

Another study that would require extensive more work. We are glad that the Editor agrees that this would go far beyond the scope of this article.

Some other minor points:

1. In their response: "We did not observe any further changes after 5 minutes. In contrast, a steady state for the T108A mutant was reached between 10-20 mins. ". I think this is important for the readers and they should have included it in the main text.

Line 126: 900 sec (15 mins).

2. In their main text Line 125, they need to cite their previous work there.

Done

Reviewer #3:

Remarks to the Author:

- The authors now state that spring constants have been converted to forces by multiplying by the amplitude of the oscillation. This means that the obtained value will depend on the applied oscillation, and thereby rises the question of why the specific amplitude of 3 μm was chosen. Is it because it fully unfolds the membrane undulations?

Also, this important clarification is mentioned in the response to reviewers but not in the text. In the text, an estimated value of 1 pN is now mentioned, without any clear explanation.

We thank the reviewer for this point worth of clarification, since it is our best interest to emphasize our rationale (in the text). This amplitude value was chosen to avoid GUV detachment (force-reference loss) due to a “harsh” oscillation. Indeed, we attempted to sweep amplitudes (e.g 1-10 μm) for each tube-data point, but without success. Irrespective of the specific amplitude, the idea was to have a “reference” for the force range required to unfold the lipid membrane reservoir from the GUV. Then the respective amplitude was applied for the FtsZ-springs. For some vesicles, the oscillation amplitude was not sufficient to fully unfold the membrane excess and then the force induced by FtsZ range falls into the lipid contribution. For others, the membrane is completely unfolded, and in this case, we are able pick up the resistive contribution from FtsZ itself.

Irrespective of the amplitude, in a very coarse approximation, there is a force signal that corresponds to a number of “ring-units” (as clearly defined in the text). 1 pN refers to the force per ring unit. In a first order approximation, this relation should be valid for different amplitudes and number of rings units.

We will update the text accordingly. The expression “membrane unfolding” has not been used in the text and it is very self-explanatory. Thanks for suggesting it.

- Regarding cross sections of the tubes, I thank the authors for clarifying their calculations.

However, can the authors clarify how the flexural rigidity is calculated from persistence length? Also, please note that in line 184 they mistakenly state that $E = KI$, the correct expression is $K = EI$.

The reviewer is correct to point a typo in line 184, and we apologize for overlooking it. In addition, the flexural rigidity K (or bending rigidity) has been calculated using:

$$l_p = \frac{K}{k_B T}$$

according to (along others)

1. Chase Broedersz and Fred MacKintosh. Modeling semiflexible polymer networks.

Reviews of Modern Physics, 86(3):995–1036, July 2014.

2. Steven S. Andrews. Methods for modeling cytoskeletal and DNA filaments. *Physical Biology*, 11(1):011001, January 2014.

We will update the text by citing these papers to provide a clear reference.

- Finally, in page 7 the authors state that they discard spring constant values of FtsZ tubes that are in the range of membrane only tubes. In my view, this is not correct: by definition, this approach will lead to differences between FtsZ and lipid tubes, even if they are not there. From the data points, it is clear to me that there are indeed differences, so this is not a major concern, but I think the authors should:

- o Quantify spring constants considering all values, the fact that some FtsZ values fall within the lipid range is to be expected considering the high variability observed in both cases.

- o Then, I think it is fine for the authors to subtract both obtained spring constants to estimate the FtsZ contribution, as they do.

Exactly, this rationale was also referred to in our comment above. FtsZ force can only be picked up in fully unfolded membranes. Thus, it is valid to discard FtsZ data that is obscured in the lipid-induced force regime.

As mentioned above, the text will be updated to make this point clearer.

REVIEWERS' COMMENTS

Reviewer #2 (Remarks to the Author):

The revision has amended the previous version with new texts and clarifications. I wish the authors to consider the following points and clarify further in their writings:

1. The non-hydrolyzable GTP analog experiment was asked to examine whether the further condensation of the lipid tubes is indeed caused by treadmilling dynamics or a structural change of the filament upon GTP hydrolysis. From the authors' responses, my understanding is that besides the fact that such an experiment is extremely difficult, the authors are not really distinguishing the difference between the treadmilling dynamics and the structural changes it causes. Those two, however, mechanistically speaking, are different. A force generated by a treadmilling dynamics implies that it is the association and dissociation processes of monomers that generate a torsional force to the membrane—when each monomer is attached to or detached from the membrane, the interaction (its associated energy) generates a force, and hence causes a deformation of the membrane. A structural change is caused by the geometry of the polymer, which can be static and maintained without any dynamics. Imagine a gymnast jumping on a monkey bar v.s. just standing on a bar.

2. To extend this analysis further, my understanding is that the authors are also not distinguishing the difference between treadmilling dynamics and GTP hydrolysis. Strictly speaking, treadmilling requires GTP hydrolysis in the FtsZ system, but GTP hydrolysis is not sufficient to generate treadmilling. To generate treadmilling, there needs to be a spatial cue: this is provided by the intrinsic polymerization polarity of FtsZ, and facilitated by membrane binding; the latter eliminates many other competing processes. Can a FtsZ polymer formed in solution treadmill? Perhaps, under certain conditions that constrain the geometry of the polymers—there are reports of overexpressed bacterial FtsZ forming toroidal structures in yeast, which, could treadmill as well. Also, there must be concentration gradients so that a FtsZ monomer can continuously add to the growing tip and dissociate from the shrinking end. Imagine if all FtsZ monomers are attached to the membrane permanently (mts, ZipA all are reversible membrane association for FtsZ) and they diffuse slowly (or the density on membrane is high), they can still hydrolyze GTP, but can they still treadmill?

3. The authors measured the force by membrane deformation, which, unfortunately, is hard to decouple from the treadmilling v.s. structural causes, because both would deform the membrane. The ideal experiment would be having two FtsZ filaments of same structures/curvatures, but one is treadmilling and the other is not. I do not know what experiment could achieve this—perhaps if there is a way to measure force without using membrane deformation, such as using non-deflated

or tense lipid and incorporate a force tension sensor between the membrane tether and FtsZ? I am not suggesting the authors to do this experiment for the current paper, but it is important to think about these different scenarios and make the texts precise.

4. The authors did provide convincing evidence to show that WT FtsZ caused further compression of the helical tubes compared to the T108A mutant, but as I explained above, these two polymers differ in both ring diameters (structural) and treadmilling behaviors (dynamics) and it is hard to decouple them.

5. Another point about the ZipA-FtsZ system: the authors believe that adding ZipA does not change FtsZ's treadmilling speed. This is true based on the recent publications, but probably only to a certain concentration ratio of ZipA/FtsZ. Imagine if all FtsZ monomers are attached to membrane by ZipA, the diffusion of FtsZ on the membrane will be different from that in the cytoplasm, and hence the on/off rate (treadmilling rate) for the addition and reduction of FtsZ monomers will be different. It looks like that the GTPase activities of FtsZ-ZipA and FtsZ-mts are different, which could be consistent with this expectation. Therefore, the authors could tune treadmilling speed using the ZipA-FtsZ system and hence further tease out the effect of treadmilling v.s. structural change.

6. I do not see a different title of the main text: it is still the same "Bidirectional treadmilling FtsZ filaments transform lipid membranes via torsional stress" compared to what I received the last time. Is it really treadmilling that transforms lipid? Or is it the structural changes of the polymer enhanced by GTP hydrolysis? Additionally, is bidirectional absolutely necessary? Or just being directional is enough? I do not think this point was proven or disproven in the revised text.

7. I agree that this work has merit, the experiments are hard and do provide new observations. However, as I articulated above, the authors are not really distinguishing the cause of the deformation of the membrane as treadmilling, GTP hydrolysis, or structural changes of the polymers caused by GTP hydrolysis. They are all highly correlated, but the mechanisms to generate a force are different. As such, it would be hugely helpful to the field if the authors could clearly spell out these differences and draw the conclusion as being consistent, but not proving, that treadmilling dynamics deform the membrane.

8. I think if the authors do not want to do the ZipA-FtsZ over expression experiments in vivo, it may make the story more coherent by leaving the last section out and use it elsewhere with proper controls. Messing up with cell division can have multiple consequences (FtsZ-mts is not in a natural state for divisome assembly), lysis and bulging are commonly encountered, and may not be necessarily specific to FtsZ-YFP-mts overproduction. L-form cells show similar morphologies; lipid

dynamics together with the entropic force from chromosome segregation can constrict L-form cells (or spheroplasts). The data presented there are descriptive and not particularly informative.

Reviewer #3 (Remarks to the Author):

The authors have adequately addressed my concerns, and in my view the manuscript is now ready for publication. I also concur with the editor in that the current in vivo evidence is sufficient to illustrate the potential implications of the findings, and that further in vivo work should be done in follow-up studies. I do have however one minor comment left. After the authors clarifications, it is clear to me that the value of 1 pN corresponds to the force f that a FtsZ ring would apply when deformed by a distance x of 3 μm (divided by the number of rings in the pulled tube), given a measured spring constant k . Smaller or larger deformations would lead to smaller or larger forces, as given in first order by Hooke's law ($f=kx$). So, to evaluate whether this force corresponds to what FtsZ rings would actually apply upon GTP hydrolysis, the authors should estimate if the order of magnitude of the associated ring deformations (upon GTP hydrolysis) would match that applied by optical tweezers. If not, the authors can simply recalculate their force estimate by multiplying their measured spring constant by the actual deformation induced by GTP hydrolysis.

I ask simply to estimate this (not even precisely, just in terms of orders of magnitude) from the data from this manuscript or from the literature, I am not requesting new experiments.

Reviewer #2 (Remarks to the Author):

The revision has amended the previous version with new texts and clarifications. I wish the authors to consider the following points and clarify further in their writings:

1. The non-hydrolyzable GTP analog experiment was asked to examine whether the further condensation of the lipid tubes is indeed caused by treadmilling dynamics or a structural change of the filament upon GTP hydrolysis. From the authors' responses, my understanding is that besides the fact that such an experiment is extremely difficult, the authors are not really distinguishing the difference between the treadmilling dynamics and the structural changes it causes. Those two, however, mechanistically speaking, are different. A force generated by a treadmilling dynamics implies that it is the association and dissociation processes of monomers that generate a torsional force to the membrane—when each monomer is attached to or detached from the membrane, the interaction (its associated energy) generates a force, and hence causes a deformation of the membrane. A structural change is caused by the geometry of the polymer, which can be static and maintained without any dynamics.

Imagine a gymnast jumping on a monkey bar v.s. just standing on a bar.

We appreciate that the problem of this reviewer with our claim of treadmilling-induced forces is that we do not have any structural insight of how exactly the force could be transmitted between FtsZ and the membrane during treadmilling. This is true, as our methods do not yield the atomic resolution that would be required. Instead, we demonstrate that GTP hydrolysis that leads to treadmilling is also connected to an – additional – force beyond a passive, geometry induced deformation, which is a necessary, but not sufficient requirement for the claim that treadmilling as such induces a force. We toned down our claim, also changing the title, to make sure that the two phenomena are linked, but not necessarily by a strict causal relationship of one resulting from the other.

2. To extend this analysis further, my understanding is that the authors are also not distinguishing the difference between treadmilling dynamics and GTP hydrolysis. Strictly speaking, treadmilling requires GTP hydrolysis in the FtsZ system, but GTP hydrolysis is not sufficient to generate treadmilling. To generate treadmilling, there needs to be a spatial cue: this is provided by the intrinsic polymerization polarity of FtsZ, and facilitated by membrane binding; the latter eliminates many other competing processes. Can a FtsZ polymer formed in solution treadmill? Perhaps, under certain conditions that constrain the geometry of the polymers—there are reports of overexpressed bacterial FtsZ forming toroidal structures in yeast, which, could treadmill as well. Also, there must be concentration gradients so that a FtsZ monomer can continuously add to the growing tip and dissociate from the shrinking end. Imagine if all FtsZ monomers are attached to the membrane permanently (mts, ZipA all are reversible membrane association for FtsZ) and they diffuse slowly (or the density on membrane is high), they can still hydrolyze GTP, but can they still treadmill?

See above.

3. The authors measured the force by membrane deformation, which, unfortunately, is hard to

decouple from the treadmilling v.s. structural causes, because both would deform the membrane. The ideal experiment would be having two FtsZ filaments of same structures/curvatures, but one is treadmilling and the other is not. I do not know what experiment could achieve this—perhaps if there is a way to measure force without using membrane deformation, such as using non-deflated or tense lipid and incorporate a force tension sensor between the membrane tether and FtsZ? I am not suggesting the authors to do this experiment for the current paper, but it is important to think about these different scenarios and make the texts precise.

We agree that this would be an interesting experiment to perform in the future.

4. The authors did provide convincing evidence to show that WT FtsZ caused further compression of the helical tubes compared to the T108A mutant, but as I explained above, these two polymers differ in both ring diameters (structural) and treadmilling behaviors (dynamics) and it is hard to decouple them.

See our answer to point 1.

5. Another point about the ZipA-FtsZ system: the authors believe that adding ZipA does not change FtsZ's treadmilling speed. This is true based on the recent publications, but probably only to a certain concentration ratio of ZipA/FtsZ. Imagine if all FtsZ monomers are attached to membrane by ZipA, the diffusion of FtsZ on the membrane will be different from that in the cytoplasm, and hence the on/off rate (treadmilling rate) for the addition and reduction of FtsZ monomers will be different. It looks like that the GTPase activities of FtsZ-ZipA and FtsZ-mts are different, which could be consistent with this expectation. Therefore, the authors could tune treadmilling speed using the ZipA-FtsZ system and hence further tease out the effect of treadmilling v.s. structural change.

See above. We modified our claims and hope that more experiments in the future will reveal further insights into this question.

6. I do not see a different title of the main text: it is still the same “Bidirectional treadmilling FtsZ filaments transform lipid membranes via torsional stress” compared to what I received the last time. Is it really treadmilling that transforms lipid? Or is it the structural changes of the polymer enhanced by GTP hydrolysis? Additionally, is bidirectional absolutely necessary? Or just being directional is enough? I do not think this point was proven or disproven in the revised text.

We have now changed the title accordingly.

7. I agree that this work has merit, the experiments are hard and do provide new observations. However, as I articulated above, the authors are not really distinguishing the cause of the deformation of the membrane as treadmilling, GTP hydrolysis, or structural changes of the polymers caused by GTP hydrolysis. They are all highly correlated, but the mechanisms to generate a force are different. As such, it would be hugely helpful to the field if the authors could clearly spell out these differences and draw the conclusion as being consistent, but not proving, that treadmilling dynamics deform the membrane.

See above.

8. I think if the authors do not want to do the ZipA-FtsZ over expression experiments *in vivo*, it may make the story more coherent by leaving the last section out and use it elsewhere with proper controls. Messing up with cell division can have multiple consequences (FtsZ-mts is not in a natural state for divisome assembly), lysis and bulging are commonly encountered, and may not be necessarily specific to FtsZ-YFP-mts overproduction. L-form cells show similar morphologies; lipid dynamics together with the entropic force from chromosome segregation can constrict L-form cells (or spheroplasts). The data presented there are descriptive and not particularly informative.

We do believe that the *in vivo* experiments, although being mostly descriptive, are sufficiently new and interesting to include them to the manuscript.

Reviewer #3 (Remarks to the Author):

The authors have adequately addressed my concerns, and in my view the manuscript is now ready for publication. I also concur with the editor in that the current *in vivo* evidence is sufficient to illustrate the potential implications of the findings, and that further *in vivo* work should be done in follow-up studies. I do have however one minor comment left. After the authors clarifications, it is clear to me that the value of 1 pN corresponds to the force f that a FtsZ ring would apply when deformed by a distance x of 3 μm (divided by the number of rings in the pulled tube), given a measured spring constant k . Smaller or larger deformations would lead to smaller or larger forces, as given in first order by Hooke's law ($f=kx$). So, to evaluate whether this force corresponds to what FtsZ rings would actually apply upon GTP hydrolysis, the authors should estimate if the order of magnitude of the associated ring deformations (upon GTP hydrolysis) would match that applied by optical tweezers. If not, the authors can simply recalculate their force estimate by multiplying their measured spring constant by the actual deformation induced by GTP hydrolysis. I ask simply to estimate this (not even precisely, just in terms of orders of magnitude) from the data from this manuscript or from the literature, I am not requesting new experiments.

Given our FtsZ spring constant measurements and ring density-per-micrometer, we could estimate the GTPase activity force contribution by using the difference in pitch from Figure 3. Then, a difference of $\sim 2 \mu\text{m}$ would account for a GTPase-associated force of 0.9 pN. This agrees with the range of forces here reported in our study, and it suggests an important contribution of the GTPase activity to the overall force.